# International Survey of Specialist Fetal Alcohol Spectrum Disorder Diagnostic Clinics: Comparison of Diagnostic Approach and Considerations Regarding the Potential for Unification

**DOI:** 10.3390/ijerph192315663

**Published:** 2022-11-25

**Authors:** Natasha Reid, Dianne C. Shanley, Jayden Logan, Codi White, Wei Liu, Erinn Hawkins

**Affiliations:** 1Child Health Research Centre, The University of Queensland, Brisbane, QLD 4072, Australia; 2School of Applied Psychology, Griffith University, Gold Coast, QLD 4222, Australia; 3Menzies Health Institute of Queensland, Griffith University, Gold Coast, QLD 4222, Australia

**Keywords:** fetal alcohol spectrum disorder, diagnostic criteria, barriers and facilitators

## Abstract

Fetal alcohol spectrum disorder (FASD) is a prevalent neurodevelopmental condition. Despite FASD being recognized as a clinical disorder there is no globally agreed set of diagnostic criteria. Accurate and timely diagnosis of FASD is imperative to inform clinical care, optimize outcomes for individuals accessing assessments and their families, as well as for research and prevention strategies. To inform movement towards a unified approach, the present study aimed to capture an international perspective on current FASD diagnostic criteria, as well as potential barriers and facilitators to unification. An online survey was created using REDCap and sent to clinics identified and contacted via internet searches. Quantitative data were presented using descriptive statistics and open-ended questions analysed using content analysis. The survey captured information about each clinic’s current diagnostic approach, whether they would support a unified method, and the barriers and facilitators for a consistent international FASD diagnostic approach. Fifty-five (37.4%) of 147 FASD clinics identified worldwide participated. The majority (*n* = 50, 90.9%) of respondents supported a unified approach. Content analysis identified a lack of collaboration as a key barrier, while strong leadership in guideline creation and implementation emerged as a central facilitator. These barriers and facilitators can be used to guide future collaborative efforts towards implementing consistent diagnostic criteria.

## 1. Introduction

Fetal alcohol spectrum disorder (FASD) is a condition that arises from prenatal alcohol exposure [1]. Individuals with FASD can experience a wide range of both physical (e.g., growth impairment, facial dysmorphology and congenital defects) and neurodevelopmental impacts (e.g., cognition, language, memory and learning impairments). These impacts can lead to increased risk of secondary conditions, including disengagement from education and increased risk of involvement with the justice system [1,2,3,4,5]. Timely intervention plays a key role in limiting secondary adverse outcomes including mental health and substance misuse conditions, disconnection from school, and impaired capacity for independent living and working arrangements [6,7]. An international systematic review and meta-analysis estimated that the global prevalence of FASD among children and youth was 7.7 per 1000 (range 4.9–11.7 per 1000) [8]. Prevalence in populations at higher risk of FASD (e.g., correctional facilities, out-of-home care) has been estimated to be 10–40 times higher than the global prevalence estimate [9]. Furthermore, the mean annual cost for children with FASD is estimated at USD $22,810, with a total lifetime cost of at least USD $2 million [10,11]. 

The absence of a biomarker or objective test to diagnose FASD has resulted in a reliance on symptom-based diagnostic criteria. While there is widespread scientific and medical recognition of FASD as a clinical disorder, no standardised international diagnostic criteria currently exists. For example, three different diagnostic approaches are currently used in the United States alone [12,13,14]. Canada has also published a guideline, which informed the development of guidelines in both Australia and Scotland [1]. There was also a guideline published in Germany and more recently, there has been criteria (Neurobehavioural Disorder Associated with prenatal alcohol exposure—ND-PAE) proposed in the Diagnostic and Statistical Manual of Mental Disorders, Fifth Edition (DSM-5) [15,16,17]. Consequently, there is no single agreed upon approach for the diagnosis of FASD. Discordance between diagnostic criteria has resulted in misdiagnoses, variability in government messaging, limits the ability to compare results between FASD studies and leads to confusion for both clinicians and individuals accessing assessments [18]. 

A previous international survey of diagnostic services for children with FASD identified 34 clinics: 20 from the survey and 14 from published literature [19]. The clinics were in North America (*n* = 29), Africa (*n* = 2), Europe (*n* = 2) and South America (*n*= 1). At the time of this previous survey, no clinics were found in Asia or Australasia. The most common criteria reported was the Washington 4-Digit Diagnostic Code (*n* = 14), followed by Hoyme et al. IOM criteria (*n* = 8), and one clinic was using the original IOM criteria [19]. 

The only previous survey of specialist diagnostic clinics was completed over 10 years ago [19]. Gathering direct information from clinicians who are the end users of clinical practice guidelines can provide vital information regarding practical facilitators and barriers to improve service provision for individuals accessing assessments. This is also a key part of the process in developing new clinical practice guidelines, as guidelines developed with input from end users leads to increased relevancy, enhanced legitimacy and improved dissemination and uptake of clinical practice guidelines [20,21,22,23]. 

The aim of the current study was to collect up-to-date information from an international perspective regarding guidelines utilised at specialist FASD diagnostic clinics. Additionally, we were interested in collecting input from clinicians regarding: (1) if they thought it would be possible and worthwhile to develop a unified international diagnostic approach; and (2) what the barriers and facilitators to developing a unified diagnostic approach would be.

## 2. Materials and Methods

### 2.1. Participants and Recruitment 

A list of potential specialist FASD diagnostic clinics was compiled through online searching (*n* = 176). Clinics were contacted via email and provided information regarding the study and a link to the online survey. Information was provided to request that one member of the clinic team complete the survey. If a clinic did not respond to the initial email, a single follow-up email and phone call were completed. Further follow up contact was not carried out due to the COVID-19 pandemic. Following initial contact 29 clinics were found to not be specialist FASD diagnostic services and were removed from further contact. Data were collected from October 2018 until July 2020.

### 2.2. Questionnaire 

An online survey was created and administered using REDCap) hosted at Griffith University, Australia [24].The survey has been included in Appendix A.

*Clinic demographic information*. The survey collected key demographic variables (e.g., location, age of clients, type of clinic). 

*Diagnostic guidelines*. One multiple-choice question asked the clinician to select which diagnostic criteria was used in their clinic. Participants could select multiple options if multiple criteria were used. An ‘other’ option was also included where participants could record any additional criteria that were used. 

*Unified approach to diagnosis*. Two yes/no questions (i.e., do you think that it is possible to develop a unified international diagnostic guideline for FASD diagnosis; do you think there is a need for a unified international guideline for FASD diagnosis) were included. Two open ended questions were included (i.e., what are the barriers to a unified international guideline for FASD diagnosis; what are the facilitators for a unified international guideline for FASD diagnosis?). 

*Assessment tools*. Fourteen additional questions were included that gathered information regarding the neurodevelopmental domains assessed and the tools currently used. To enable comprehensive discussion of the current and assessment tool results a separate study will be written to provide a summary of the assessment tool results. 

### 2.3. Data Analysis

Quantitative data were summarised using descriptive statistics, whereby categorical variables were reported as frequencies and percentages. Open-ended questions were analysed using content analysis. Content analysis is used to study responses to open-ended questions through utilisation of counting and coding written text into groups and patterns [25,26]. Responses were examined to identify unique barriers and facilitators, each identified code was assigned a label. Multiple codes could be identified within a single response (e.g., barriers of ‘cost’ and ‘time’); however, each code was only counted once per individual (e.g., ‘expensive’ and ‘funding needed’). After initial codes were developed across the dataset by one author (CW), individual codes were re-examined by the research team to further refine the larger overarching themes. Where agreement was reached, these were collapsed to provide the final identified barriers and facilitators. 

## 3. Results

### 3.1. Clinic Demographics and Criteria Usage

From the 176 contacted clinics, 29 were excluded as ineligible as they were not providing assessment and diagnostic services for FASD resulting in 147 potentially eligible clinics. Of the 147 invited clinics, 60 were in the USA, 50 in Canada, 10 in New Zealand, 16 in Australia, 3 in the United Kingdom, and 8 in other nations. From the eligible clinics, fifty-five (37.4%) completed the survey. The clinics which completed the survey comprised the following proportions of initially invited nations: USA (37%), Canada (20%), New Zealand (80%), Australia (50%), United Kingdom (100%), Other (50%). Table 1 provides a summary of the key demographic information for the clinics. Table 2 provides a detailed summary of the current diagnostic criteria that were used in the specialist clinics that participated in the survey. The majority of participants (*n* = 50; 90.9%) who completed the survey reported that a unified diagnostic system would be possible and that it would be worthwhile (*n* = 49; 89.1%). Responses that included multiple guidelines (*n* = 17) have been summarized in Table A1. 

### 3.2. Barriers to a Unified Diagnostic Approach

Clinicians in the current sample identified significant barriers to the development of a unified diagnostic approach (Figure 1; *n* = 49). These barriers aligned with four overarching themes: lack of collaboration, lack of generalisability, problems with criteria development, and implementation issues. The most common overarching theme was lack of collaboration, that is problems with either the likelihood of collaboration or the collaboration process of creating a unified criteria. This included the most common theme of individual or institutional egos, personal preference, or tradition in using past diagnostic guidelines (*n* = 19). Other process barriers included lack of agreement (*n* = 8), cost/lack of funding (*n* = 7), lack of unified leadership (*n* = 4) and poor communication (*n* = 3). The second overarching theme was lack of generalisability, that is problems with a one-size-fits-all approach. This theme included barriers in terms of the diversity of settings, with lack of ability to standardise due to setting differences in resources, training and specialist access (*n* = 10) and also due to perceived differences in diagnostic purpose across different settings (*n* = 3). There were also concerns that a single set of guidelines may be too rigid (*n* = 1), and that there would be difficulty in operationalising or adapting a unified diagnostic guideline across different cultural groups and regions (*n* = 4) and across different age groups (*n* = 2). The third overarching theme was problems with criteria development, that is problems with the content of a unified diagnostic guideline. The largest concern was that there was a lack of empirical data (*n* = 8) to support a unified diagnostic guideline. There were specific concerns that a) existing inconsistencies in terminology or diagnostic criteria would hinder unification (*n* = 8), and b) disagreement over the need for and method of confirming prenatal alcohol exposure (*n* = 4). The final overarching theme was implementation regarding concerns about adoption of a unified guideline. Specifically, barriers were identified regarding policy and prescribed use of certain guidelines by funding bodies (*n* = 4). Given that some of these themes only included a low number of responses, the discussion will focus on the most representative data.

### 3.3. Facilitators to a Unified Diagnostic Approach

Clinicians who participated in the present survey also identified a number of factors that could facilitate the development of a unified diagnostic approach (Figure 1; *n* = 38). Facilitators reported by participants were grouped into four overarching themes: guideline creation and implementation, stakeholder characteristics, stakeholder groups, and guideline characteristics (existing and needed). The most common overarching theme was guideline creation, that is methods of organising the unification. This included the most common individual theme: presence of a strong objective leadership, collaboration, steering committee or international organising body to facilitate unification (*n* = 9). Other methods identified as facilitators included accurate empirical evidence or testing of guidelines (*n* = 5), sharing research findings (*n* = 4), surveying current guideline use (*n* = 3) and networking or communication (*n* = 4). Similarly, funding (*n* = 2), political support (*n* = 2) and recognition of the importance of FASD internationally (*n* = 2) were identified as potential facilitators for the creation and implementation of unified diagnostic guidelines. The next overarching theme was stakeholder characteristics. Specifically, participants reported a need for involved parties to have open mindedness (*n* = 5), having an interest or demand for consistency (*n* = 3) and a desire to help families (*n* = 3) would facilitate unification. The third overarching theme involved who comprised the stakeholder groups. Participants reported that clinician involvement (*n* = 4) or researchers-clinician collaboration (*n* = 3) would help facilitate unification, while some reported that involvement of a younger generation might be needed (*n* = 3). The last overarching theme involved characteristics of the guidelines themselves. Specifically, some participants identified that there was a need to simplify guidelines and diagnosis (*n* = 3), while others noted that the existing similarity across guidelines could facilitate unification (*n* = 2). Given that some of these themes only included a low number of responses, the discussion will focus on the most representative data.

## 4. Discussion

### 4.1. Diagnostic Criteria

The present study aimed to capture an international perspective on current FASD diagnostic criteria, as well as discern potential barriers and facilitators to taking a unified approach. The results included responses from fifty-five identified specialist clinics across North America, Oceania, and Europe. From these responses, more than ten diagnostic criteria were found to be in use. Importantly, most clinics surveyed employed diagnostic strategies originating from guidelines developed in their respective nation or region. For example, 9/10 Canadian respondents solely use Canadian guidelines, while 7/8 Australian clinics applied Australian guidelines. This finding points to a tendency for clinics to prefer methods developed within unique clinical, geographic, and cultural settings. Despite this apparent partiality towards contextualised guidelines, 90.9% of respondents supported the need for and possibility of a future universal diagnostic strategy. 

Limitations of currently used methods has been a key driver for the emergence of new approaches. A previous study that applied five different diagnostic tools to 1581 patients found moderate agreement (Cohen’s κ coefficient 0·24–0·58) between compared measures [27]. While some of the criteria had considerable overlap (i.e., growth features), others showed significant discrepancies (i.e., neurobehavioural and physical features) [27]. The result of these differences ultimately led to a wide range of individuals diagnosed with FASD (4.74–59.58%) based on the criteria applied [27]. A second, more recent study reported similar discordance in diagnostic outcomes when researchers applied the 4-Digit Code, Canadian 2015, Australian 2016 and Hoyme 2016 diagnostic guidelines to 1392 patient records [28]. This study found that the key factors contributing to differences include: high alcohol exposure, specific facial criteria, inclusion/exclusion of growth and brain criteria, and exclusion of moderate dysfunction [28,29]. These studies highlight that while clinics seem to be using varying criteria under the broad context of FASD diagnoses, each criterion cannot be universally applied in the same way. Considering current diagnostic criteria used today, key differences can be summarised as the inclusion of growth deficits as a criterion, the number of sentinel facial features and cut-off of palpebral fissure length required for diagnosis, process for assessing prenatal alcohol exposure, and conceptualisation and clinical cut-offs applied in neurodevelopment [1,12,14]. 

On top of the differences between diagnostic criteria, individual clinician interpretation of specific guidelines has also been shown to influence diagnoses, with a tendency for inclusion of borderline cases [30]. Accurate and timely diagnosis of FASD is imperative to inform clinical care, as well as for research and prevention strategies. It is essential to consider the implications for over and under diagnoses of FASD when considering a future universal approach [31]. Differences between current criteria, introduction of new criteria, and adaptation of existing methods all have the potential to cause significant harms and burdens to patients diagnosed, their families, healthcare systems, and research outcomes [18]. With respect to research findings, diagnostic inconsistencies are a key driver of variation in FASD prevalence statistics and government action that follows reporting [18]. On the other hand, past studies have shown that a diagnosis of FASD can be empowering for individuals and their caregivers [32,33]. Accurate diagnosis facilitates improved understanding of the condition, whilst providing avenues to seek and obtain appropriate support services [33]. Whilst updates to current guidelines should be approached cautiously, there is still clearly room for improvement. A recent study provided information regarding the current revision and updates to the Australian Guide to Diagnosis of FASD, based on the perspectives of clinicians, individuals with lived experience and cultural expertise, and our evolving understanding of the disorder [22]. Importantly, this study captured that on top of guideline content, dissemination and implementation were both priority areas that need addressing in the review process [22]. This body of research considered together with our finding of majority support highlights the need for a universal diagnostic strategy.

Hospital and community agency clinics utilised similar proportions of individual FASD guidelines, while private practices reported higher rates of multiple guidelines. Given the aforementioned variation in FASD diagnosis based on guidelines utilised, it is difficult to determine how higher rates of multiple guidelines may affect specific FASD diagnoses. These results do however illustrate that a significant proportion of FASD diagnoses are being made in the private sector. It is therefore important to consider how clinic type may affect uptake and implementation of a future universal diagnostic.

### 4.2. Barriers to a Unified Diagnostic Approach

When considering barriers to a unified diagnostic approach, four overarching themes were identified from the current study including: lack of collaboration, lack of generalisability, lacking specific diagnostic criteria, and implementation/adoption issues. The most commonly reported specific barriers were personal preferences, tradition in use of specific diagnostic guidelines. Complexity and heterogeneity of FASD presentations resulted in a divergence in the way different clinical/research groups approach diagnosis [34]. This effect has been compounded by existing gaps in research regarding FASD development, and a lack of context-specific guidelines, which have together hindered an evidence-based approach. This lack of consensus led many specialist FASD clinics around the world to develop and adopt their own set of guidelines in isolation [35]. Together these factors have contributed to the belief captured in this study that personal preferences, ego, and tradition will be a key barrier to a universal diagnostic approach. More broadly, our findings were similar to other studies that have previously reported barriers to the implementation of clinical practice guidelines [36,37,38,39]. A scoping review of the broad barriers to clinical guideline implementation found that clinician knowledge and awareness, attitudes towards guidelines, staffing, and program sustainability were key barriers to enactment [37]. To begin to understand how to overcome these barriers, it is important to recognise where they stem from. Many factors have been shown to influence attitudes and therefore, behaviours regarding practice guidelines. Lack of motivation, inertia of previous practice (habitual/routines), challenge to autonomy, lack of self-belief, and difficulty in understanding and agreeing with guidelines, have all been previously reported to influence physician guideline uptake [38,40]. Development and implementation of a unified guideline would require attitude and behavioural shifts from researchers and healthcare professionals, as well as adjustment of structural environments [37].

One of the current challenges with applying specific diagnostic criteria is the subtlety of elements that each criteria requires. An absence of historical information on alcohol exposure, delayed onset and/or identification of cognitive and behavioural deficits, and marginal differences in physical features complicate many cases [41]. Some of the most recent attempts to update guidelines, the Revised IOM and Revised Canadian Diagnostic Guideline methods, try to overcome this with reliance on improved specialist clinical teams as a key method [1,14]. It is important to acknowledge that the emergence of new evidence along with clinical experience has allowed for more sensitive diagnoses [41]. However, this has also contributed to greater divergence among diagnostic approaches. This is essential to consider for a future universal approach. Some settings simply do not have the resources and specialist access (reported in Figure 1), which may influence capacity for implementation. Research has shown that clinicians require specialist training to improve rates and capacity of accurate FASD diagnosis, particularly for recognition of physical features [42]. While this highlights education as a key area to improve future detection, it also reinforces a lack of resources as a potential barrier. Additionally, it should be noted that discrepancies in diagnostic criteria are consistent with a lack of a clear definitions in the wider neuropsychology and anthropometric disciplines with regard to cut-offs for impairments [43,44,45,46]. This sentiment was reinforced by our findings that a lack of empirical evidence is a key barrier to diagnostic consensus. The field could benefit from an up-to-date study that provides a detailed analysis of the current diagnostic criteria and the empirical evidence used to inform the development of the criteria. Such a study may benefit from using the Appraisal of Guidelines, Research and Evaluation (AGREE)-II to appraise the quality of reporting of the diagnostic guidelines. This would provide a first important step in clarifying both applicability and feasibility for moving current methods towards a future universal approach.

The variation in applied diagnostic criteria captured by the present study highlights that there is a lack of consensus on which guideline provides the best diagnostic outcome across settings. A recent study evaluated the usefulness of commonly utilised FASD guidelines in a Polish context. The researchers concluded that mainstream methods could not be applied in Poland without significant adaptations taking into account the Polish health service, education, and social structures [35]. The study also emphasised that within Poland a lack of consensus and oversight has meant a tendency to work in isolation, furthering divergence between diagnostic strategies [35]. Again, this highlights the need for an up-to-date study comparing current diagnostic approaches and how applicable they may be across settings. It may be that a core set of diagnostic criteria could be agreed upon at an international level and then individual countries or regions could develop their own local guidelines to inform the implementation of the diagnostic criteria across different contexts, informed by local health service strengths and needs. 

Other key facilitators identified by the current study were open communication, open mindedness, simplification of guidelines, and approaches based on empirical evidence. Studies have shown that clear communication between management and professionals, with clearly defined responsibilities, is a key facilitator for the implementation of clinical practice guidelines [36]. Our findings are further reinforced by others who reported that interventions based on consistent and clear clinical evidence are most likely to have successful outcomes [40]. This study also reported that for complex interventions, it is essential for health care professions and governing organisations to consider an array of contextual factors [40]. This point is particularly important given the aforementioned complexities to FASD diagnoses and the diversity of presentations.

### 4.3. Facilitators to a Unified Diagnostic Approach

Respondents in our study identified numerous facilitators for the development of a unified diagnostic approach. Responses were captured under four broad themes including: guideline characteristics (existing and needed), key stakeholder groups, stakeholder characteristics, and guideline creation/implementation methods. From these themes, strong leadership along with well-defined empirical evidence emerged as central. Leadership is an essential facilitator for development and implementation of clinical guidelines [36,47]. It underpins clear objectives, facilitates action and collaboration, and fosters enthusiasm [47]. This necessity of strong leadership to successful guideline implementation warrants the question: Who is an appropriate governing body for universal FASD diagnostic criteria? Responses from the present study suggest a steering committee facilitated by an international body. The World Health Organisation’s International Classification of Diseases (ICD), and the American Psychiatric Association’s Diagnostic and Statistical Manual of Mental Disorders (DSM) are the two main classification systems for conditions [48]. Both the ICD and DSM are developed and co-authored by hundreds of subject matter experts (clinicians and researchers), overseen by relevant governing bodies. Importantly, both systems include perspectives on an international scale, place emphasis on empirical research, and publish in numerous languages. A recent study comparing the ICD-11 and DSM-5 found significant overlap in terms of the disorders described as well as a general shift towards harmonisation (between ICD and DSM) from previous versions [48]. This study supports the notion that irrespective of which of these systems may be used, future publications are likely to converge in their diagnostic descriptions. It should be noted that the DSM-5 has received criticism for a lack of transparency in its development, and tendency to over-pathologise normal psychological distress [49,50]. Together these criticisms emphasise the importance of both a transparent review process, as well as rigorous appraisal of empirical evidence in the development of new diagnostic guidelines. One method to facilitate this task is the ‘Grading of Recommendations Assessment, Development, and Evaluation’ (GRADE) approach [51]. GRADE can be used to assess the certainty of the evidence and applicability to the future goal of a universal FASD diagnostic strategy [51]. Utilising a GRADE approach would involve considering the best available evidence along with potential benefits and harms, resources and costs, feasibility, and acceptability across settings. 

Notably, four respondents of the present study reported already using the DSM-5 as primary diagnostic approach. Neurobehavioral disorder associated with prenatal alcohol exposure (ND-PAE) is a term listed under ‘conditions for further study’ in the appendix of the DSM-5 [17]. ND-PAE is a classification that can be applied to a broad range of neurobehavioural effects, which may or may not meet the criteria for fetal alcohol syndrome [52]. This further demonstrates the need for convergence of diagnostic approaches. 

### 4.4. Strengths and Limitations

A strength of the current study is that it captured responses from clinics utilising a wide range of diagnostic criteria. This variation allowed for comparisons to be drawn from clinics operating using unique guidelines and provided insights into experience/context-specific barriers and facilitators. In capturing shared barriers between settings, we have provided priority areas that can be used to inform future approaches to a universal diagnostic approach. There were also several limitations that should be acknowledged, particularly those related to recruitment of participants. We received a relatively low respondent rate (37.4%). The precise reason for a lack of response from many clinics is not clear, however it may be attributed to factors such as lack of time to participate. While we did follow-up with a single email and a single phone call, further follow-up was ceased due to the COVID-19 pandemic. Future studies may improve response rates from clinic by implementing more extensive follow-up measures. These more extensive measures would have been appropriate given the expected schedule of practitioners.

The method of identifying relevant clinics was online searching. This approach means that we excluded clinics without websites, which may mean a bias towards higher resourced settings. Internationally, future research and practice could be improved through the establishment of an international registry of diagnostic clinics to facilitate research and communication between clinicians worldwide in the FASD field. Furthermore, most responses came from the United States, Canada, Australia, and New Zealand. This may be due in part to our correspondence being sent only in English or may be reflective of the relative concentrations of worldwide specialist clinics. As such, it is difficult to say which FASD guidelines are being used outside of these regions, and what barriers and facilitators clinicians and researchers in these settings foresee to a universal approach. Given that the results include a higher proportion of clinics in New Zealand, the United Kingdom, and Australia than were eligible, these nations are overrepresented in responses. Another limitation is that we did not capture information about the role of the specific respondent. 

There is also the potential for selection bias in that the clinics were more likely to respond if they saw the benefit of a future unified approach. Although feedback from these clinics is still important, engaging a larger number of clinics and clinicians who do not support a unified diagnostic could have provided unique insights regarding potential barriers and facilitators in these contexts. Given this, future studies would benefit from exploring further the reasoning why some respondents did not support unification. 

## 5. Conclusions

A unified FASD diagnostic criteria is necessary to standardise global disease management, clinical care, and research outcomes. Discrepancies among the many currently employed criteria has emerged from a tendency to work in region-specific isolation, insular research and clinical practice, and a lack of a governing body to facilitate consensus. This study highlights that there is clear support for a universal diagnostic approach. A focus on up-to-date empirical evidence, strong leadership, openness, and inclusion, are key facilitators for any effort towards unification. Ultimately, implementing consistent diagnostic criteria will enable improved patient outcomes, and better collaborative research efforts.

## Figures and Tables

**Figure 1 ijerph-19-15663-f001:**
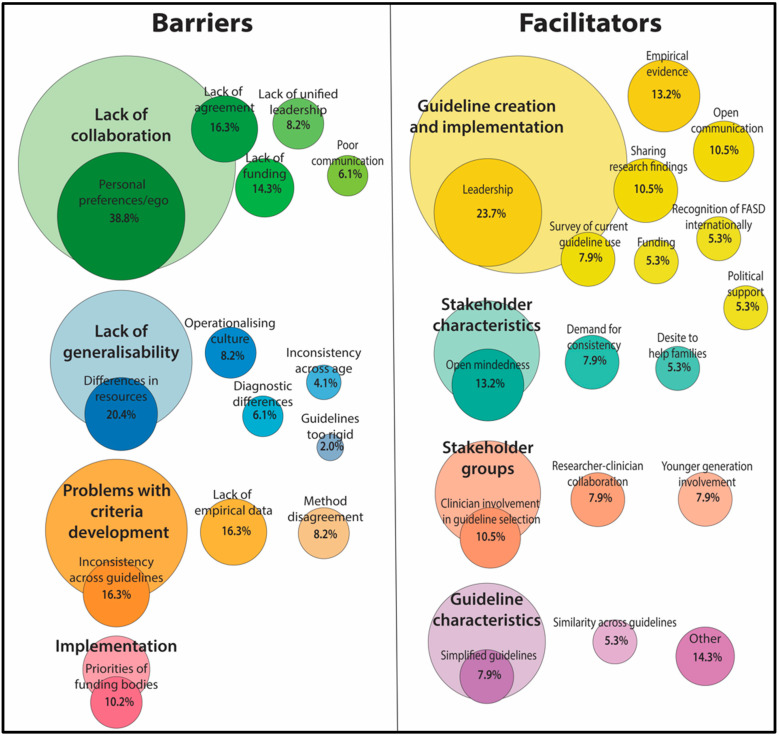
Barriers and facilitators to a unified FASD diagnostic criteria.

**Table 1 ijerph-19-15663-t001:** Clinic demographics and use of three mostly commonly used guidelines.

	Clinic Frequency	Canadian Guidelines (2016) (*n* = 15)	Australian Guidelines (Bower & Elliot, 2016) (*n* = 7)	4 Digit Code (Astley, 2004) (*n* = 12)	Other Guidelines (*n* = 4)	Multiple Guidelines (*n* = 17)
**Country**						
United States	22			10	4	8
Australia	8		7			1
New Zealand	8	6				2
Canada	10	9				1
UK	3					3
Other	4			1	1	2
**Clinic Type**						
Hospital	20	7	3	3	3	4
Community agency	14	6	3	4	0	1
University	6	0	0	2	2	2
Private practice	15	1	1	3	0	10
Other *	6	2	1	1	0	2

Note: *, e.g., not-for-profits, contracted to government, part community/part private. Other included Austria, Germany, Poland & Italy these locations have been grouped together so data are not identifiable. Total numbers provided for criteria here do not include the multiple guideline options, which are counted separately.

**Table 2 ijerph-19-15663-t002:** Diagnostic criteria implemented.

Diagnostic Criteria	Frequency
Australian Guide to FASD Diagnosis (Bower & Elliott, 2016)	9
Canadian Diagnostic Guideline (Chudley et al., 2005)	3
Revised Canadian Diagnostic Guideline (Cook et al., 2016)	22
Revised United States Institute of Medicine Criteria (Hoyme et al., 2005)	6
4-Digit Diagnostic Code (Astley, 2004)	21
Centers for Disease Control and Prevention (Fetal Alcohol Syndrome: Guidelines for Referral and Diagnosis, 2004)	7
Emory Clinic Diagnostic Guidelines	2
DSM 5	4
Revised United States Institute of Medicine Criteria (Hoyme et al., 2016)	2
Other	7

Note. Frequency reported here includes the multiple options, i.e., participants could select multiple options. Other options included 1996 IOM Guidelines, FAS/ARND diagnostic checklist, German Guidelines, SIGN Guidelines.

## Data Availability

The data presented in this study are available on request from the corresponding author. The data are not publicly available due to ethical considerations.

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
