# Peer review of "International Survey of Specialist Fetal Alcohol Spectrum Disorder Diagnostic Clinics: Comparison of Diagnostic Approach and Considerations Regarding the Potential for Unification"

_ijerph, 2022, doi:10.3390/ijerph192315663_

Round 1
Reviewer 1 Report
This study proposed to address the following issue by conducting a survey of FASD clinic personnel. The authors note that despite FASD being recognized as a clinical disorder there is no globally agreed set of diagnostic criteria. Accurate and timely diagnosis of FASD is imperative to inform clinical care, as well as for research and prevention strategies. To inform movement towards a unified approach, the present study aimed to capture an international perspective on current FASD diagnostic criteria, as well as potential barriers and facilitators to unification. An online survey was created using REDCap and sent to clinics identified and contacted via internet searches. Quantitative data were presented using descriptive statistics and open-ended questions analyzed using content analysis.”
The topic is central to the clinical field of FASD. Over the decades, numerous FAS/D diagnostic guidelines have been published, often specific to a particular country or region. When FAS was first identified and coined in the U.S. by Jones et al., (1973), the diagnostic criteria defining the syndrome were understandably qualitative (gestalt) in nature, awaiting guidance from more thorough clinical and research efforts to better understand and more specifically case-define the condition. Attempts to refine the clinical criteria for FAS/D appear in the published clinical literature (Clarren & Smith, 1978; Rosett 1980; Sokol & Clarren, 1989) leading up to the seminal publication commissioned by the Institute of Medicine (IOM, 1996). Each of these refinements had one thing in common, they all used a more qualitative (gestalt) approach to diagnosis. For example, growth deficiency has been a key diagnostic criteria for FAS across all FASD diagnostic guidelines since 1973. but none of the guidelines provided specific criteria (case definitions) for what constituted growth deficiency (e.g., below the 3rd percentile? Below the 10th percentile?; prenatal and/or postnatal deficiency?; reduced height for weight; etc.). The lack of case-definitions for the different components of FAS (growth, face, brain, alcohol exposure) resulted in extremely high variability and inaccuracy in diagnostic outcomes both between clinicians and within a single clinician. The first attempt to case-define the criteria for FASD was introduced with the 4-Digit Code in 2004. The 4-Digit Code was unique in that it was published as an empirical study confirming it’s performance was clinically and statistically superior to the 1996 IOM guidelines it was intended to replace. Establishing an empirical evidence-base has long been a standard of clinical and research practice for moving a field forward. Interestingly and unfortunately, all FASD diagnostic guidelines established worldwide subsequent to the 4-Digit Code (Canadian, Australian, Revised IOM, Polish, Scottish, German, etc.) were established and published without the benefit of empirical analyses to confirm the new guideline’s performance was superior to existing guidelines. Most were released without ever having been used in a clinical setting. They were often established by a committee charged with reviewing the existing literature and reaching consensus on what criteria to maintain, alter or eliminate. The resulting new criteria are never validated (subjected to empirical analysis). And as anyone knows who has been a member of a consensus committee, compromise is required to achieve consensus, but compromise does not always align with science. This is the primary obstacle that has prevented the field of FASD from moving forward toward a unified, validated, evidence-based approach to FASD diagnosis. Requiring diagnostic guide authors to test the performance of their guidelines on a large, representative clinical sample and confirm the performance of their new proposed FASD diagnostic guidelines are superior to existing guidelines prior to publication would prevent the endless release of new, unvalidated FASD diagnostic guidelines.
The current study contacted clinician from FASD diagnostic clinics worldwide to see if they thought it would be possible and worthwhile to develop a unified international diagnostic approach and what they believed were the barriers and facilitators to developing a unified diagnostic approach would be. I can understand why one might think this is the population to target for such a survey, but in all honesty, clinicians typically do not select the FASD diagnostic guidelines they use, their regional medical policies (e.g., medical insurance agencies, hospital networks, legislative mandates) and research agencies like NIH do. Typically, they are required to use a particular diagnostic system to qualify for reimbursement for their services or obtain a federal research grant. It is highly unlikely the clinicians are truly informed about what the strengths and weaknesses are of the diagnostic system they are using. It is even less likely that they have an accurate understanding of how the different diagnostic systems compare. This is due in part to the less than accurate publications comparing the performance of different diagnostic systems (addressed below). And many clinicians would not have the necessary study design and statistical training to critically review the evidence base (or lack of evidence base) supporting the diagnostic system they are using. This comment is not meant to be insulting to clinicians. It simply reflects the fact that clinicians rarely have sufficient training in research methodology and statistics to accurately interpret the empirical evidence. They are excellent clinicians, but that does not make them excellent researchers. Assessing the accuracy, precision and validity of a diagnostic tool is an area of expertise in and of itself.
Strengths
The results are interesting and concisely presented, but as expressed below, I am concerned that the outcomes are likely misleading due to the challenges encountered to locate and enroll a representative sample of clinicians. It is also not entirely clear how these survey responses facilitate or inform the creation of a unified diagnostic system.
Limitations
A major limitation of the study is the potential that the outcomes of the study are inaccurate and nonrepresentative due to the challenges the authors encountered locating their target population (clinicians from specialist FASD diagnostic clinics worldwide) and surveying those who were located. The authors note this limitation, but simply stating it as a limitation does not overcome the adverse impact the limitation has on the validity of the results. When investigators conduct a study, it is incumbent on them to confirm the study population is sufficiently representative of the target population to generate valid data. Based on the following, it appears highly unlikely that this study population is sufficiently representative of the target population to generate valid results. First, there is no directory of all FASD clinics worldwide, thus the author’s establishment of a list of 147 FASD clinics worldwide through an internet search is unlikely to generate a comprehensive or representative sample. Second, the survey was in English only, which may account for the near complete absence of European, African, Russian, Indian, Nordic, and South American FASD clinics that do exist. Finally, of the 147 clinics located predominantly in the US, Canada, Australia and New Zealand, only 55 (37%) participated. When there is such a large discordance between the study population and target population, it is incumbent on the authors to provide evidence that the study population, while small, is at least representative of the target population. We already know the study population is not representative of the worldwide clinics. But are the 55 clinics reasonably representative of the 147 eligible clinics? In Table 1, is the distribution of countries across the 55 clinics equivalent to the distribution of countries across the 147 clinics.
The online search for FASD specialist clinics only identified 176 worldwide of which 29 were deemed not FASD specialist clinics. The 29 excluded clinics need to be better defined. What made them ineligible?
FASD clinics typically include multiple clinicians. Did more than one clinician from each of the 55 clinics participate in the survey? If yes, this needs to be made clear in the manuscript because it would invalidate the numbers presented in Tables 1 and 2. If only one clinician per clinic was allowed to participate in the survey, which clinician was it (the director, the MD, the social worker, etc)? This needs to be clarified throughout the manuscript.
In Table 1, when multiple guidelines were used, the authors should report which multiple guidelines were used.
For reasons explained above, the statement “The most commonly used criteria was the revised Canadian Guideline, following closely by the 4-Digit Diagnostic Code and then the third most commonly used (albeit at a lower rate than the first two) was the Australian Guide to FASD diagnosis” is inaccurate and should be deleted. The study sample is so nonrepresentative of worldwide or even English-speaking countries, the authors have no ability to conclude which diagnostic guidelines are most commonly used. Even the title of Table 1 inappropriately infers that the Canadian, 4-Digit Code and Australian guidelines are the 3 most commonly used guidelines. They are the most commonly used in this nonrepresentative study sample, but Readers could misinterpret this to mean they are the most commonly used across the target population of all FASD clinics worldwide. If all 147 clinics had participated in the survey, the most commonly used guidelines among the 147 surveyed may be the Australian Guidelines for all one knows. Clearly the guidelines most likely to be used in the US, Australia and Canada are the US, Australian and Canadian guidelines respectively. So, if the study enrolled a higher proportion of Australian clinics than US or Canadian clinics, it would look like the most common guidelines used were Australian guidelines. This is the key reason why publishing a study with a highly nonrepresentative sample can result in Readers misinterpreting the outcomes. If all 147 clinics had participated in this study, all outcomes reported in this study could be significantly different than what is being reported for the subset of 55 clinics that chose to participate.
I do not understand how to interpret the data presented in Table 2 and how it relates to (or if it relates to) data in Table 1. Table 2 reports 9 participants reported use of the Australian guidelines. But in Table 1, the header under Australian Guidelines is (n = 7). Please clarify.
The title of the paper “International survey of specialist fetal alcohol spectrum disorder diagnostic clinics: comparison of criteria and considerations regarding the potential for a unified diagnostic approach” implies a comparison of criteria was conducted. I interpret “criteria” to mean the actual criteria used by each guideline. It would be more accurate to state a “comparison of diagnostic approach” was conducted.
The abstract fails to convey the nonrepresentative study sample. The statement “From the eligible clinics, fifty-five responses were received.” needs to be revised as follows: “Fifty-five (37.4%) of 147 FASD clinics identified worldwide participated.”
In section 4, the authors discuss two publications that assessed the level of diagnostic disagreement when multiple FASD diagnostic guidelines were applied to a single patient population. One study assessed 5 different diagnostic tools (revised IOM 2005; 4-Digit 2004; Canadian 2005; CDC 2004; Emory). Another assessed FAS between two tools (IOM2005; 4-Digit 2004). Both studies assessed outdated FASD diagnostic tools and one study had a significant methodological flaw warranting an Erratum. A more comprehensive and current publication comparing the 4 key diagnostic systems represented in this manuscript should be referenced (Astley Hemingway SJ, Bledsoe JM, Brooks A, Davies JK, Jirikowic T, Olson EM, Thorne JC. Comparison of the 4-Digit Code, Canadian 2015, Australian 2016 and Hoyme 2016 fetal alcohol spectrum disorder diagnostic guidelines. Advances in Pediatric Research 2019 6:31. doi:10.35248/2385-4529.19.6.31). The study not only illustrated the striking discordance in diagnostic outcomes between the 4 diagnostic systems, it also identified the 5 key factors that accounted for the greatest contrasts in diagnostic outcomes. This is the type of information that is needed to inform the development of a single valid diagnostic system that will meet the needs of all stakeholders, most importantly the families impacted by FASD.
Author Response
Please find attached our responses.

Reviewer 2 Report
From line 31, please change the sentence as follows:
“Individuals with FASD can experience a wide range of both physical (e.g., growth impairment, facial dysmorphology and congenital defects), neurodevel- opmental (e.g., cognition, language, memory and learning impairments) [1] and criminal impacts (behavioral problems, developmental delay, attention deficit hyperactivity disorder (ADHD), alcohol abuse involvement with the criminal justice system) (Notes*)
*Banerji A., Shah C., Ten-year experience of fetal alcohol spectrum disorder; diagnostic and resource challenges in Indigenous children. Paediatr. Child Heath, 22, 143–147, 2017.
*Brownell M., Enns J.E., Hanlon-Dearman A., Chateau D., Phillips-Beck W., Singal D., MacWilliam L., Longstaffe S., Chudley A., Elias B., et al. Health, Social, Education, and Justice Outcomes of Manitoba First Nations Children Diagnosed with Fetal Alcohol Spectrum Disorder: A Population-Based Cohort Study of Linked Administrative Data. Can. J. Psychiatry, 64, 611–620, 2019.
* Sessa F., Salerno M., Esposito M., Di Nunno N., Li Rosi G., Roccuzzo S., Pomara C., Understanding the Relationship between Fetal Alcohol Spectrum Disorder (FASD) and Criminal Justice: A Systematic Review, Healthcare, Jan 2, 10(1),84, 2022, doi: 10.3390/healthcare10010084.
* Kambeitz, C., Klug, M.G., Greenmyer J., Popova S., Burd L., Association of adverse childhood experiences and neurodevelop-mental disorders in people with fetal alcohol spectrum disorders (FASD) and non-FASD controls. BMC Pediatr, 19, 498, 2019.
Author Response
Please find attached our responses.

Reviewer 3 Report
Review for ijerph-1960844International survey of specialist fetal alcohol spectrum disorder diagnostic clinics: comparison of criteria and considerations regarding the potential for a unified diagnostic approach by Reid et al.
General comment:
This paper is extremely important for today clinical practice regarding diagnostic of the complex FASD disease. Despite this general positive comment, I have many major concerns which preclude the acceptance of the paper in its present form.
I have major comments notably on the analysis performed and the way the data are presented and consequently in the discussion part.
Major comment
1) In general, the number of questions is very low and this can be a limit of the study. Is there any justification for such a low number of questions?
2) Is it possible to know more about the type of people who answered the survey: physicians, pediatric doctors, addictologists, nurses maybe or others since the responses may vary accordingly. This point needs to be taken into account.
3) One limitation is of course the low number of responding clinics (37.4%) and this should be clearly mentioned although the reasons of such a low figure is never clear, but hypothesis should be given to explain this low percentage. Maybe also a proposal to improve that percentage should be presented by the authors
4) The authors present the clinic type. However, it is unclear why this is important because the authors did not cross their data with this parameter. Any explanations or precision to make?
5) The authors present the client age range (patient age?) without exploiting this criterion in the responses given. What is then the need of the client age range in the present paper?
6) The way the overarching themes were selected is rather subjective and do not fit with a so called “identification” of themes. Indeed, and as specified in the methods, overarching themes were selected by the research team after a discussion leading to a consensual agreement between the member of the team. Thus, there is no clear identifiers given or used. Although this is understandable by nature, it is difficult to read “identification” of themes in such conditions. Please, change the vocabulary.
7) Regarding the results about barriers and facilitators, the authors tried to be exhaustive in their analysis; however, this leads to disparate responses without clear message for the reader. One possibility would be to clearly indicate the objectives of this type of analysis. What were exactly the scientific questions in this context?
8) The way these data are presented do not allow the reader to check the work of the authors. On one hand, the text gives some names to the responses that do not fit with those used in the figure 1. On the other hand, some percentages given in figure 1 do not fit to the n= given in the text (see for example “flexibility” at 2.4% while in the text it is given as n=1/49 “too rigid”, which is 2.04%). See also the words “lack of agreement” in the text that become “lack of consensus” in the figure…or “bodies” in the text that became “priorities of funding bodies” in the figure…
9) Some n= are missing in the results such as “problems with criteria development”. This n should be indicated.
10) What means reporting some n=1 or 2 in the responses, that is 2.04% or 5.4%?. These are rather anecdotic n. In other words, how could we expect this kind of analysis to help solving the present problem? Again, what were the objectives?
11) Same comments (5, 6, 7, 8) for the facilitators.
12) The authors should try to represent their results with a table indicating the n= and the corresponding % instead of using circles that do not bring anything else than disturbances.
13) Discussion: the paper is about diagnostic of FASD and the first paragraph somehow conclude that the most universal criteria used are those for FAS. This needs to be strongly pointed out since FASD do not necessarily show the strong physical signs of FAS indicating the high difficulty to diagnose FASD with the present available tools, leading possibly to some mistake in diagnosis.
14) In this first part, the need to also inform the family needs to be mentioned as families have strong guilty feelings when confronted to a lack of diagnosis on their children.
15) This first paragraph of the discussion should be more linked to the results of the present study.
16) The second paragraph about barriers needs to mention that one solution may be to strongly consider (or to remind to the reader) FASD as a spectrum disorder and not an all or nothing disorder. In this context, the use of cut-off in diagnosis become less important. Also, the education of medical doctors needs to be emphasized as one possible solution for the future detection and thus diagnosis of FASD.
17) The third paragraph do not include enough discussion about the present result about the facilitators found, especially when compared to other studies in the field.
18) Does the necessity of a strong leadership a unique and specific response to the FASD diagnosis problem?
19) A conclusion needs to be done in what new brings the present study compared to previous one in the field of FASD diagnosis
20) 90% of responders agreed for a unified diagnostic system and considered it is feasible. It is thus surprising that the authors conclude that there is a need to incorporate those who do not support a unified system. What about improving the number of responding clinics as a first step to know better the feelings/the need of the clinicians confronted to FASD?
Mino comments
Abstract:
21) The words “common...condition” are not that appropriate when taking about FASD. FASD cannot be (and should not be) considered as “common” since this word or expression trivialize the dramatic situation of a series of neurodevelopemental alterations induced by a specific prenatal ethanol exposure.
22) The diagnosis is also important for families, not only for practicians
23) Please, give the number of “eligible clinics” obtained in the survey
Introduction:
24) The word “clients” is somehow difficult to read while thinking about “patients” or “subjects” (line 55)
25) Line 62. A reference is needed.
26) Line 63-64: there seems to be a confusion in here as the authors mentioned their present interest in the criteria used to diagnose FASD whereas there is no such a list in the article. At most, the authors selected the kind of guidelines in use in the different clinics
Material and methods:
27) Line 79: The country of the Griffith Univ should be mentioned
28) Line 82: change diagnostic criteria for diagnostic guidelines that seem more appropriate
29) Could we have any information about the time needed to answer the survey?
Result
30) It seems obvious that due to the low number of responses, the analysis ended up with some items being very poor in terms of number of responses. Some are only n=3 or even n=1. Are they really worth mentioning? This again call for limitations of the study because of the low number of clinics that responded to the survey.
31) The above comment is even more pronounce regarding the “Facilitators”, rendering difficult a main and strong message to be delivered to the reader.
Author Response
Please find attached our responses.

Round 2
Reviewer 1 Report
The manuscript has been improved, but there remain some significant concerns.
The methods section does not adequately convey what questions were on the survey. The authors cite Harris et al, 2009, but do not include that citation in the reference list.
The tables (1, 2 and S1) reporting the number of clinics using the different FASD diagnostic guidelines are very difficult to interpret (and at times uninterpretable). I cannot get the numbers in Tables 1 and S1 to match the numbers in Table 2. I can get the Australian guidelines frequency in Table 2 to equal 9 by adding the Australian guideline total in Table 1 (n = 7) to the Australian guidelines total in Table S1 (n = 2). When I take the same approach to derive the Frequency for the 4-Digit Code in Table 2 (n=21) I get n = 23 (e.g. 12 from Table 1 and 11 from Table S1), not 21 as reported. If I add the 15 Canadian guidelines from Table 1 to the 4 revised Canadian guidelines from Table S1, I match the 22 Revised Canadian guidelines in Table 2. But the 2005 Canadian Guidelines should be 2 not 3 as reported in Table 2. What makes these tables so difficult to reconcile with one another is the authors do not clearly report to Readers which guidelines they are reporting (e.g., are the Canadian Guidelines in Table 1 the 2005, 2016, or both)? The authors also inaccurately label the IOM guidelines in the column heading of Table S1. There is only one set of FASD guidelines commissioned by the IOM. They were published in 1996. The Hoyme et al 2005 and 2016 FASD guidelines are revisions of the IOM guidelines. These revisions were not commissioned by the IOM and thus should not be labeled as such.
The authors response to Point #2 is inadequate. The major limitation of the study remains a major limitation. A standard of clinical research methodology is to document how representative the study population is relative to the study's target population. The authors did not provide or report this requested information. The revised text in the limitations section is a step in the right direction, but it remains unclear why the authors do not report what proportion of invited clinics were from each country. 147 clinics were invited to participate. Only 55 (37%) participated. If all 147 participated, the results of the study could be strikingly different if the 55 are not representative of the 147. The authors report "the most commonly used criteria, from responses captured in the present study, was the revised Canadian Guidelines , following closely by the 4-Digit Diagnostic Code and then the (albeit at a lower rate than the first two) was the Australian Guide to FASD diagnosis." This statement is meaningless (and apparently inaccurate based on my calculations above; Revised Canadian 22, 4-Digit 23) if the study sample was not representative of the 147 targeted clinics. For example, if the majority of the 147 clinics that did not participate were Australian, the data would likely have shown the Australian guidelines were the most commonly used. Since the authors have not provided the requested information on the 147 clinics, they should delete the above statement that implies which guidelines are most commonly used from the manuscript.
Author Response
Please find attached our responses.

Reviewer 3 Report
Review 2 for ijerph-1960844
International survey of specialist fetal alcohol spectrum disorder diagnostic clinics: comparison of criteria and considerations regarding the potential for a unified diagnostic approach by Reid et al.
General comment:
In general, the authors properly answered most of my queries but I still have few major corrections to ask before the paper can be fully accepted.
Major points:
1) Regarding the low number of responding clinics (37.4%) which is now acknowledge as a limitation of the study by the authors, it is noted in the revised version that such a low figure can be attributable to lack of time of the practitioners to participate to the survey. However, such “hypothesis” could be easily counteracted if we would have the duration of the questionnaire to answer. A data I wished to obtain but that the authors are unable to give. In the same vein, the follow-up consisted in ONE single email and ONE phone call, which look rather low in regard of the expected schedule of practitioners. I thus suggest the authors to change the new revised sentence taking into account the present comment. Indeed, one sentence about the present methodology can be added indicating the survey was difficult to perform because of the COVID-19 pandemic and that it should be improved in a future study.
2) I am afraid I do not understand the need to present the type of clinics answering the survey and the age of the patient’s population concerned by the survey, as the authors do not cross these data with the responses they collected. The present study is not a clinical study per se in which we are used to present demographic data about the population studied. In addition there is no study of the patients here but of the practitioners and we would have liked to have the demographic data of such a population. However, I agree that the clinic type is an interesting data and that a word should be say in the discussion part about that. Otherwise, this is useless data.
3) Thus, regarding the age of the patients. Do the authors believe these data interacted with the responses to the questionnaire? If yes, how can it be? If no, what is the rationale of presenting the age of the patients in the present scientific questioning?
4) The authors accepted my requirement about making a table out of their data instead of the circle presentation for which I clearly know the advantages. However, one main rule is to avoid duplicating presentation of the same data set in a single paper. The authors should then be able to choose the way they want to present data. It is either table or circle presentation, but both should be strongly avoided even if presented as supplemental figures.
5) If the authors choose the circle presentation, a way to improve the figure would be to align the largest circles (first column on the left side of “barriers” and “facilitators”) on their respective diameter since these are the main results. Alignment will avoid the present image looking more like a dispersion of results than transmitting a clear and main message.
6) Regarding the report of very low percent of responses in “barriers” and “facilitators”. I understand that the authors are looking for transparency and completion of the results. I also understand that they emphasized on the main results. However, it should be stress in the result section that such low figures are not representative of the majority of responses and/or that they are minority opinion in the responses collected.
No minor corrections asked
Author Response
Please find attached our responses.
